# Knowledge and Perceptions of Adverse Events Following Immunization among Healthcare Professionals in Africa: A Case Study from Ghana

**DOI:** 10.3390/vaccines7010028

**Published:** 2019-03-08

**Authors:** Peter Yamoah, Varsha Bangalee, Frasia Oosthuizen

**Affiliations:** 1Komfo Anokye Teaching Hospital, Okomfo Anokye Road, Kumasi 00233, Ghana; 2College of Health Sciences, University of KwaZulu Natal, Durban 4041, South Africa; bangalee@ukzn.ac.za (V.B.); oosthuizenf@ukzn.ac.za (F.O.); 3Department of Pharmacy Practice, Faculty of Pharmacy and Pharmaceutical Sciences, Kwame Nkrumah University of Science and Technology, Accra Rd, Kumasi 00233, Ghana

**Keywords:** vaccine pharmacovigilance, Vigibase, healthcare professionals, adverse events following immunizations, vaccine safety

## Abstract

The spontaneous reporting of suspected adverse events following immunization (AEFI) by healthcare professionals (HCPs) is vital in monitoring post-licensure vaccine safety. The main objective of this study was to assess the knowledge and perceptions of AEFIs among healthcare professionals (HCPs) in Africa, using the situation in Ghana as a case study. The study was of a cross-sectional quantitative design, and was carried out from 1 July 2017 to 31 December 2017 with doctors, pharmacists, and nurses as the study participants. A 28-item paper-based questionnaire, delivered by hand to study participants, was the data collection tool in the study. The study was conducted in 4 hospitals after ethical approval was granted. The desired sample size was 686; however, 453 consented to partake in the study. Data were analyzed using SPSS (software version 22, IBM, Armonk, NY, USA), and chi-square and binary logistic regression tests were used for tests of association between HCPs’ characteristics and their knowledge and perceptions. Detailed knowledge of AEFIs was ascertained with a set of 9 questions, with 8 or 9 correctly answered questions signifying high knowledge, 5 to 7 correctly answered questions signifying moderate knowledge, and below 5 correctly answered questions signifying low knowledge. A set of 10 questions also ascertained HCPs’ positive and negative perceptions of AEFI. Results revealed that knowledge of AEFIs was high in 49 (10.8%) participants, moderate in 213 (47.0%) participants, and low in 191 (42.2%) participants. There was no statistically significant correlation between AEFI knowledge and professions. The highest negative perception was the lack of desire to learn more about how to diagnose, report, investigate, and manage AEFI, whereas the lowest was the lack of belief that surveillance improves public trust in immunization programs. There was a general awareness of AEFIs among HCPs in this study. However, negative perceptions and the lack of highly knowledgeable HCPs regarding AEFIs were possible setbacks to AEFI diagnosis, management, prevention, and reporting. More training and sensitization of HCPs on AEFIs and vaccine safety will be beneficial in improving the situation. Future research should focus on assessing the training materials and methodology used in informing HCPs about AEFIs and vaccine safety.

## 1. Introduction

Vaccines unarguably play a very key role in the prevention of infectious diseases across all populations of the world [1]. There are several types of vaccines used in specific populations such as children, adolescents, pregnant women as well as the general adult population, and their application may differ between jurisdictions [2]. In Ghana, for instance, immunization for children less than five years old is targeted at preventing 12 infectious diseases: tuberculosis, poliomyelitis, diphtheria, pertussis, tetanus, hepatitis B, *Haemophilus influenza* type b (Hib), measles, rubella, pneumonia, rotavirus, and yellow fever. These vaccines are administered at various stages of the development of the children. This immunization schedule begins with the Bacillus Calmette Guerin (BCG)/tuberculosis vaccine and the oral polio vaccine (OPV), which are given to the neonate within 24 hours of delivery, followed by diphtheria, pertussis, tetanus, Hib and hepatitis B combined penta (5-in-1) vaccine, pneumococcal vaccine (PCV), OPV, and rotavirus vaccines at 6 weeks of age, 5-in-1 vaccine, PCV OPV, and rotavirus at 10 weeks of age, 5-in-1 vaccine, PCV, and OPV at 14 weeks of age, measles and rubella combined vaccine and yellow fever vaccine at 9 months of age, after which the measles vaccine is given at 18 months of age [3]. Aside from the standard immunization schedule, booster doses of these vaccines are given as and when they are needed during infectious disease outbreaks. In the UK, 12 vaccines are also administered to children below 5 years of age, with three of their vaccines (meningitis B, meningitis C and mumps vaccines) not part of the schedule used in Ghana. In place of these three vaccines used in the UK, Ghana has BCG, tetanus, and yellow fever vaccines. In the UK infant immunization schedule, diphtheria, tetanus, pertussis, poliomyelitis, Hib and hepatitis B vaccines are given in a single combined formulation (6-in-1 injection), in addition to PCV, rotavirus vaccine, and meningitis B vaccine for infants aged 2 months, as their first vaccines after birth. This is followed by a second dose of the 6-in-one vaccine and rotavirus vaccine at age 3 months. Thereafter, a third dose of the 6-in-1 vaccine and second doses of PCV and meningitis B are given. Finally, a fourth dose of Hib and a first dose of meningitis C vaccine, given as a single formulation, measles, mumps, and rubella (MMR) as a single formulation and third doses of PCV and meningitis B are administered between 12 and 13 months of age [4]. Even though vaccines are very potent preventers of infection, they are associated with adverse reactions, most of which are minor, yet of global health concern. For instance, the BCG vaccine, which is administered to about 100 million newborns across the world annually, is associated with a few adverse events such as fever, rash, pain, suppurative adenopathy, anaphylaxis, and abscess at injection site [5,6].

Healthcare professionals (HCPs) form a very significant group among the stakeholders of vaccine safety [7]. Vaccines go through the phases of drug development from discovery, preclinical testing, and clinical trials just like pharmacological agents do, before approval by regulatory agencies for use [8,9]. Post-market surveillance of vaccines after their regulatory approval is crucial, because it helps to identify rare and late-occurring adverse events that were not discovered during clinical trials. The spontaneous reporting of suspected adverse events following immunization (AEFI) by HCPs is vital for the monitoring of post-licensure vaccine safety [10]. Evidence shows that in most low and middle-income countries (LMICs) without robust pharmacovigilance systems, HCPs play a significant role in observing medicine related harms and documenting them; this process has often led to the improvement in the functionality of pharmacovigilance systems, or their establishment in countries without these systems [11]. Africa lags behind in the field of vaccine pharmacovigilance, as revealed by an analysis of data in the World Health Organization (WHO) vaccine safety database, Vigibase, in June 2015. This analysis revealed that less than 1% of all global AEFI reports were from Africa. The analysis further showed that about 97% of these African AEFI reports came from 10 African countries, which were Egypt, Democratic Republic of Congo, Morocco, South Africa, Sierra Leone, Zimbabwe, Tunisia, Ghana, Nigeria, and Senegal [12]. This is a serious situation, as Africa has 54 countries, and more vaccine doses are being administered on the continent due to the emergence of serious infections and a burgeoning population [13,14,15]. Consequently, more AEFIs are expected than the handful reported by the few African countries observed in the Vigibase analysis above. It is therefore important for HCPs in Africa to have knowledge about AEFIs, in order to properly diagnose and report them, enhancing vaccine safety and the functionality of pharmacovigilance systems. Additionally, knowledge of AEFIs enhances treatment of life threatening conditions associated with some AEFIs [10].

The success of immunization programs depends partly on HCPs, who serve as important stakeholders in both the facilities where they work and the communities in which they live [16]. Good knowledge of vaccines and their associated adverse events could boost vaccination coverage, as HCPs will impart the knowledge they have acquired to patients in their facilities and people in their various communities, thereby improving vaccine safety and public trust in immunizations [17]. Poor knowledge and perceptions of HCPs about storage and administration of vaccines, as well as the AEFIs possible with vaccine products, have been found to be associated with failures of several immunization programs [18,19]. Vaccines lose their potency when stored at temperatures below and above the cold chain temperature range of 2 degrees Celsius to 8 degrees Celsius [20]. Apart from losing their potency through poor cold chain maintenance, temperature fluctuations could cause vaccine products to change to toxic forms, which could cause AEFIs [21,22]. For instance, in May 2017, 15 children died of severe sepsis and toxicity from contaminated measles vaccines in Kapoeta, South Sudan, due to poor vaccine cold chain storage [23]. To prevent such tragedies from recurring, HCPs must monitor vaccines routinely to ensure that they are always strictly stored within cold chain temperatures [24]. HCPs must also acquaint themselves with vaccine administration techniques in order not to inflict unnecessary pain on vaccinees, and for vaccine products to elicit their desired immunological response. Poor vaccination techniques are also a potential cause of AEFIs and must be avoided by HCPs. The WHO recommends that for children less than 15 months of age, the deltoid injection site (upper arm) should not be used when administering the inactivated polio vaccine because of the inadequate muscle mass at that age [25]. The injection of lean and undeveloped infant deltoid muscles could lead to pain, injury and abscess at the injection site, which are all classified as AEFIs; injection into the thigh muscle is therefore preferred. Moreover, the vast majority of vaccines are administered via the intramuscular route, and best vaccination practice requires that the site of intramuscular administration is stretched if the vaccine is given as an injection dosage form, in order to reduce pain, redness, and the risk of abscess formation after injection [17,26]. Furthermore, HCPs must be educated to stop certain practices which also reduce the efficacy of vaccines, as the lack of the desired effect of a vaccine is a serious vaccine pharmacovigilance issue. A classic example is the administration of analgesics such as ibuprofen and paracetamol before and around the time of administration of vaccines, with the intention of reducing pain, fever, and inflammation associated with immunization [27]. Studies have shown that such practices could interfere with the body’s antibody response to the antigenic agent in the vaccines, reducing their ability to prevent the multiplication of the disease causing organisms being targeted [28]. However, post vaccination administration of anti-pyretics, according to a systematic review, reduces fever associated with vaccination and does not interfere with the antibody response needed for disease prevention [29]. An exception to this principle is the recommendation to give paracetamol soon after immunization with meningitis B vaccine at 2 and 4 months of age in the UK, rather than waiting for a fever to develop [30]. This is done because the fever associated with the meningitis B vaccine can be very serious, and can even trigger convulsions [31].

Apart from improving knowledge of HCPs on AEFIs, negative perceptions towards AEFI reporting must also be done away with. Such negative perceptions include, but are not limited to, believing that reporting an AEFI such as injection abscess will make them feel guilty about having caused harm and been responsible for the event, and believing that combining core clinical duties with AEFI reporting is impossible [17]. Poor knowledge and negative perceptions among HCPs cut across several fields of healthcare. These have often resulted in negligence of duty which has contributed significantly to many medical liability issues [32,33]. Therefore, aside from improving AEFI reporting, high knowledge and positive perceptions about vaccination and AEFIs could help limit immunization negligence and associated medical liabilities.

With the continent of Africa being among the poor regions of the world, reducing its disease burden through wider immunization coverage could save its scarce resources tremendously, making resources available for developmental projects [34]. Wide vaccine coverage necessitates efforts to mitigate the negative effects of vaccines and immunization exercises in general. Strengthening the pharmacovigilance systems of the African continent is therefore very important in improving AEFI reporting and vaccine safety. Even though African countries such as Swaziland and Mauritius have joined the WHO Pharmacovigilance member countries in the last few years and are at the teething stages in the development of pharmacovigilance, other countries, including South Africa, Tunisia, and Ghana, have been member countries for at least the past 17 years, and have functional national pharmacovigilance systems [35]. Ghana particularly has a very solid medicines regulatory system, and stands tall among the countries in Africa with rapid development of vaccine pharmacovigilance systems. Ghana therefore serves as an example for other countries in Africa in the nascent stages of pharmacovigilance, or with no pharmacovigilance systems in place to emulate. This study is therefore being conducted in Ghana to ascertain the knowledge and perceptions of healthcare professionals of AEFIs, which will serve as a current indicator of the strength of the vaccine pharmacovigilance system in Ghana. The findings could be used as a framework for policy formulation to help improve the knowledge and perceptions of HCPs as well as their commitment to reporting AEFIs and enhancing vaccine safety in Ghana and Africa at large.

### 1.1. Study Aim

The aim of the study was to assess the knowledge and perceptions of AEFIs among HCPs in Ghana, with the hope that the findings would improve AEFI awareness elsewhere in Africa.

### 1.2. Study Objectives

To assess the AEFI knowledge of doctors, pharmacists, and nurses in Ghana.To assess the AEFI perceptions of doctors, pharmacists, and nurses in Ghana.To determine the relationship between knowledge of AEFIs and study participants’ characteristics.To determine the predictors of perceptions towards AEFIs among study participants.


## 2. Methods

### 2.1. Study Design

The study was of a cross-sectional quantitative design, carried out from 1 July 2017 to 31 December 2017.

### 2.2. Study Settings

The study was conducted in 4 hospitals in Ghana. The 4 hospitals were selected to ensure that all levels of care, i.e., primary, secondary, and tertiary, were fairly represented in the study participants. One of the hospitals was a primary healthcare (PHC) facility, two were secondary healthcare facilities (SHC), and one was a tertiary healthcare (THC) facility. A PHC facility provides health services for villages and less endowed areas in the community, with a limited presence of medical specialists. A SHC facility is more advanced than a PHC facility and is usually located in bigger towns and peri-urban centers, with some presence of medical specialists to whom cases beyond the scope of PHCs are referred. A THC facility has both medical specialists and consultants in various clinical specialties, and serves as a referral center for both PHC and SHC facilities. Two of the hospitals were situated in the northern belt, and the remaining 2 hospitals in the southern belt. The hospitals in the northern belt were the Komfo Anokye Teaching Hospital (KATH), a 1200 bed THC facility in Kumasi, and the Agogo Presbyterian Hospital (APH), a 200 bed SHC facility, both in the Ashanti region. Those in the southern belt were the Tema General Hospital (TGH), a 300 bed SHC facility in the Greater Accra region, and the Nigale Hospital (NH), a 70 bed PHC facility in the Sekondi-Takoradi metropolis of the western region.

### 2.3. Subjects/Participants

Subjects in this study were frontline clinical healthcare professionals, i.e., doctors, pharmacists, and nurses in the selected hospitals.

### 2.4. Inclusion/Exclusion Criteria

Doctors, pharmacists, and nurses in the selected hospitals who consented to partake in the study were included. Other clinical healthcare professionals apart from these were excluded from the study. Additionally, participants who had worked in the various institutions for less than 6 months were also excluded.

### 2.5. Sampling

The populations of doctors, pharmacists and nurses in the various hospitals as at the time of the study were as follows: KATH, 1200; APH, 240; TGH, 500; and NH, 30. Assuming a margin of error of 5%, a confidence level of 95%, and a response distribution of 50%, the desired sample sizes for the various hospitals based on their staff populations were as follows: KATH, 292; APH, 148; TGH, 217; and NH, 29, yielding a total sample size of 686. A list of the doctors, pharmacists, and nurses at each of these institutions was used for sampling. Sampling was done based on the proportions of the various cadres of these HCPs against the desired sample population obtained from the sample size calculations for the various institutions, as shown in Table 1 below. Arithmetic progression was used to sample the desired number of HCPs in each of the hospitals. After the arithmetic progression, 211 doctors, 25 pharmacists, and 450 nurses, yielding a total of 686, was the desired sample size for the study. These HCPs were recruited sequentially at each of the study sites until the desired sample size was reached.

### 2.6. Data Collection Tool

A structured paper questionnaire was the data collection tool in this study. The tool was prepared by the principal investigator, co-investigators, a public health physician, and a senior nurse at the immunization center of one of the study sites, the Komfo Anokye Teaching Hospital. Moreover, reference was made in the design of the data collection tool used in a similar study by Masika, Atieli and Were, conducted in Kenya, in the eastern part of Africa, and published in 2016 [17]. The questionnaire was divided into 3 sections. The first section comprised 6 questions, which were used to collect socio-demographic data, i.e., age, gender, profession, rank, years of practice, and area of practice of participants. There were 12 questions at the second section, which were used to collect data on the AEFI knowledge of participants. Out of these, 1 question assessed general awareness of AEFIs by asking about the definition of AEFI, whereas 9 questions assessed detailed knowledge of AEFIs. The third section comprised 10 questions, which were used to assess the AEFI perceptions of respondents. Altogether there were 3 open-ended questions and 25 closed-ended questions, with all the questions assessing detailed knowledge and perceptions of AEFIs being closed-ended. The rationale for including more closed-ended questions was to enhance response rate, and also to make the analysis easier.

### 2.7. Pilot Study

The data collection tool was pretested using 3 participants from each of the professional groups, including 1 participant at the various immunization centers within the hospitals. This was done to ascertain the average time taken by respondents to complete a questionnaire, the legibility, clarity, and understanding of study participants. It was discovered from the pilot study that each patient information sheet and questionnaire took an average of 13 min to complete. Data collected was analyzed to observe trends in response to the various questions. The content of the questionnaire was then restructured to improve clarity and understanding of prospective study participants, by addressing all ambiguities and anomalies observed through the responses in the pilot study.

### 2.8. Data Collection

Data was collected by one assigned research assistant in each of the hospitals. Prospective participants were contacted at their wards, consulting rooms, operating theatres, and immunization centers during working hours (8 a.m. to 5 p.m.) of Monday to Friday of the study period. Prior to administering the questionnaires, each prospective study participant was given a participant information and consent (PIC) form to complete. The PIC form contained the title of the research, name and affiliation of principal investigator, study background and purpose, study procedure, risks and benefits associated with study, voluntariness, and confidentiality of participant data. Participants who needed further clarification and explanations on the PIC form were offered this verbally by the research assistants. It was ensured that participants had written their names and signed their signatures on the designated portions of the PIC forms, after fully comprehending the contents on the forms and agreeing to partake in the study. Participants were given the questionnaires to complete immediately after the consent process.

### 2.9. Data Analysis

Data was entered and coded periodically into Epi-Data statistical software, after which it was exported to Statistical Package for Social Sciences, IBM^®^ SPSS version 22 (SPSS Inc., IBM, Armonk, NY, USA) for cleaning and analysis. Descriptive statistics were computed to generate frequencies, means, and standard deviations. General knowledge/awareness of HCPs of AEFIs was ascertained by asking the meaning of the acronym ‘AEFI’. Detailed knowledge of AEFIs was then determined using the set of 9 questions described in Section 2.6. These questions focused on vaccine product storage, vaccination technique, causes, management, diagnosis, reporting of AEFIs, and prevention and treatment of coincidental conditions following vaccine administration, and were designed such that a ‘yes’ answer was the correct one for each item. Scoring rubrics on AEFI knowledge were determined by the principal investigator and other experts of vaccine pharmacovigilance, as no study was identified in the literature with standard scoring rubrics. A mark of 1 was allotted to each correct answer, and therefore, answering all 9 questions correctly fetched a respondent 9 marks. Participants who scored 8 or 9 marks were deemed to have high knowledge of AEFIs, those scoring 5 to 7 marks were deemed to have moderate knowledge of AEFIs, and those scoring below 5 marks were deemed to have low knowledge of AEFIs. The number of study participants answering ‘yes’ to each of the AEFI knowledge questions was tallied. A two tailed chi-square test was used for tests of association between knowledge of AEFIs and participant characteristics, and *p* < 0.05 was considered statistically significant. Moreover, the set of 10 questions (Section 2.6) was used to ascertain the HCPs’ perception of AEFIs, with ‘yes’ answers being positive perceptions and ‘no’ answers being negative perceptions. A binary logistic regression test was used to determine associations between the perception of participants (positive or negative) and participant characteristics, and *p* < 0.05 was considered statistically significant. The summaries of results were presented in tables.

### 2.10. Ethical Considerations

Permission was sought from the authorities of the various hospitals where the study was conducted. After permission was granted, ethical clearance was obtained from the Committee on Human Research Publication and Ethics (CHRPE) of the School of Medical Sciences, Kwame Nkrumah University of Science and Technology (KNUST), Kumasi (Ethical approval number: CHRPE/AP/220/17).

## 3. Results

### 3.1. Socio-Demographic Characteristics of Study Participants

In this study, consent was sought from a population of 686 potential participants. Out of this number, 453 consented, completed, and returned their questionnaires, representing a response rate of 66%. The mean age of study participants was 30.6 years (SD = 6.6 years), with minimum and maximum ages of 20 years and 59 years respectively. Among the categories of HCPs there were junior staff (1–5 years working experience), senior staff (more than 5 years but neither specialists nor consultants), specialists (participants who had passed a health professional college membership examination), and consultants (participants who had passed a health professional college fellowship examination). All the specialists and consultants were doctors. The other socio-demographic characteristics of the study participants are summarized in Table 2.

### 3.2. Knowledge of AEFI

‘AEFI’ was correctly defined by 83.4% of the study participants. The top three AEFI knowledge scores were associated with the reporting of: injection site abscesses, the possible cause of AEFI by reconstituted vaccine stored longer than normal, vaccine reaction, inappropriate route of administration, vaccines stored beyond expiry date or contaminated vaccines, and treatment of a coincidental illness falsely attributed as a vaccine reaction without delay. The mean AEFI knowledge score was 5.9 (SD = 1.3), with minimum and maximum scores of 1 and 9 respectively. The summary of the knowledge of study participants of AEFIs is presented in Table 3.

### 3.3. Perceptions of AEFI

The highest positive perception about AEFIs was associated with the belief that enhancing surveillance of AEFIs can help build public trust in immunization programs, whereas the lowest was the belief that no time can be made to report AEFIs because of busy work schedules. The highest negative perception was the lack of desire to learn more about how to diagnose, report, investigate, and manage AEFI, whereas the lowest was the lack of belief in surveillance improving public trust in immunization programs. The perceptions of study participants of AEFIs are summarized in Table 4, with ‘yes’ answers indicating positive perception.

### 3.4. Relationship between Socio-Demographic Characteristics and Knowledge of AEFIs

The relationship between knowledge of AEFIs and study participant characteristics was statistically significant for work location, gender, and number of AEFI trainings within a year. The results of the chi-square tests of association are summarized in Table 5.

### 3.5. Logistic Regression of Positive Perception of AEFIs with Study Participant Characteristics

Binary logistic regressional analysis of the positive perceptions of AEFIs with study participant characteristics showed that staff working at KATH and APH had a higher positive perception of AEFI than the mean by about 3- and 5-fold respectively. Moreover, the female gender had about a 2-fold positive perception increase over their male counterparts. Additionally, HCPs in the senior rank had a 3-fold increase in positive perception about AEFIs compared to those in the junior rank. Furthermore, HCPs working at immunization clinics had about a 7-fold increase in positive perception of AEFIs over their colleagues not working in immunization clinics. Lastly, HCPs who received one and two AEFI trainings within the past year had about a 5- and 7-fold positive perception increase of AEFIs respectively, compared to their colleagues who received no training. The summary of the logistic regressional analysis is shown in Table 6.

## 4. Discussion

This study is among only a few conducted on the African continent on knowledge and perceptions of AEFIs among HCPs to include doctors, pharmacists, and nurses, as most of studies have focused on nurses alone [17,36].

Over four-fifths of study participants had knowledge of the meaning of the acronym ‘AEFI’. This is an indication of possible awareness about AEFIs among the HCPs. However, based on the scores, detailed knowledge of AEFIs was lacking; only about 10% of study participants were highly knowledgeable about the details of AEFIs, with a vast majority having low to moderate knowledge. High knowledge of the clinical manifestation of AEFIs and their prevention and treatment is beneficial in assuaging the effects of AEFIs during mass vaccine campaigns and routine immunizations, particularly of children below five years of age [37,38]. This promotes public trust in vaccines, leading to improved vaccine coverage and reducing the burden of infectious diseases as more people get vaccinated. The low knowledge finding in this study was also observed in a similar study conducted in Albania [39]. Contrary to the low detailed knowledge of AEFIs found in this study, findings of a similar study conducted in Australia showed very high knowledge of AEFIs among HCPs [40]. This could be due to the fact that most developed countries, including Australia, are committed to educating their HCPs on AEFIs, compared to the practice in most LMICs. This is, however, disputed by the fact that in a study among HCPs in Europe, only 35% of 26 studied European Union countries had developed training protocols for their HCPs on the identification, prevention, and treatment of AEFIs [41]. In Ghana, the FDA has intensified the training of both public sector and private sector HCPs on pharmacovigilance over the past few years, in the bid to improve knowledge of the safety of healthcare products, including vaccines. Additionally, the WHO Africa Collaborating Centre for Advocacy and Training in Pharmacovigilance in Ghana has been well equipped over the past 5 years to train HCPs in Ghana and the entire African region in best pharmacovigilance practices, including vaccine pharmacovigilance. Despite these interventions, the current study shows that only 20% of study participants had received official training on AEFIs within the past year. Additionally, there was a statistically significant correlation between the institution an HCP worked at and knowledge of AEFIs, according to the results of tests of association. These indicate the need to intensify training efforts and extend pharmacovigilance training to various regions of the country, to improve coverage of HCPs. A 10-month prospective study conducted in Ghana observed a 600% increase in the reporting rate of AEFIs by HCPs following training, monitoring visits, and provision of AEFI reporting forms to HCPs, lending credence to the need for training intensification [42]. This is further supported by the relationship between the number of AEFI trainings per year and HCPs’ knowledge of AEFIs, which was statistically significant in the current study. The study also pointed out that pharmacovigilance, and therefore vaccine safety knowledge, is not seriously imparted to doctors, pharmacists and nurses during their professional training in universities and colleges. Over 80% of study participants received no training in vaccine safety in school; future studies should therefore consider assessing the training curricula of these professionals, to investigate the extent to which pharmacovigilance is integrated into their educational curricula.

About 30% of the study participants had interest in investigating and reporting AEFIs. Several factors, such as the cumbersome bureaucratic process of reporting AEFIs, complex data collection tools for AEFIs, and inability to report AEFIs because of busy work schedules, could account for this lack of interest, as has been observed in vaccine and infectious disease research in other African countries [36,43]. This was complicated by just a quarter of respondents being willing to learn how to diagnose, report, investigate, and manage AEFIs, even though a vast majority believed in the positive impact of AEFI reporting on building public trust in immunization programs. In mitigating these problems, steps must be taken by regulatory authorities and managers of immunization programs to make reporting easier for HCPs, while ensuring all vital AEFI information is still collected. Other methods, such as using modern technology, could make AEFI reporting easier. For instance, one study has observed that online computer applications enhance the reporting of AEFIs more than paper forms, which were considered cumbersome by many of the study participants [44]. These web-based computer programs could be developed and simplified as smart phone applications, to make reporting easier and more convenient for HCPs by way of reporting with their smart phones. To give HCPs the confidence to report AEFIs, fears of being blamed and victimized for causing the AEFI must be allayed through sensitization workshops. According to a Zimbabwean study, key among the reasons for negative perception and low reporting of AEFI is the fear of being blamed and victimized after reporting the AEFI [45]. Results from the current study showed that about three-quarters of study participants believed that reporting an AEFI can lead to such personal consequences; such thoughts and beliefs must therefore be discarded in order to improve AEFI reporting. 

The results of the binary logistic regression indicate that HCPs in TGH and NH, the male gender, HCPs in the junior ranks, and HCPs not working in immunization clinics may need more sensitization to improve their positive perceptions about AEFIs. Gender has not been associated with positive attitude to AEFI reporting in other studies on the African continent [17,37]. It will therefore be appropriate for future researchers to confirm this finding in other study sites in Africa. However, higher education and higher ranks of HCPs have been associated with positive perceptions towards AEFIs in other studies, as observed in the current study [46,47]. As higher ranks within the health industry have mandated managerial tasks, which are associated with extra commitment alongside their core clinical duties, their positive perceptions probably emanated from these managerial duties. An unusual finding of the binary logistic regression test is the fact that there was no significant difference in positive AEFI attitude above 2 trainings per year. This could be attributed to the fact that only the first 2 trainings addressed the needs of the HCPs based on the training material, methodology, and the resource persons employed. This shows that though quantity of training is important, the quality must not be overlooked.

### Strengths and Limitations of Study

A major strength of this study is the fact that the study sites consisted of primary, secondary, and tertiary care facilities, making the sample representative of all levels of health care. There are times that vaccine cases are referred from primary healthcare settings to secondary and tertiary facilities for life threatening AEFI management, showing that HCPs’ AEFI knowledge should cut across all levels of care. A major limitation of the study is the exclusion of private healthcare facilities as study sites. Future studies are recommended to include private healthcare facilities, as some vaccines like BCG are given to neonates in these facilities within 24 h of birth, requiring AEFI monitoring by HCPs. This will make the study sample even more representative. Another limitation of the study is the fact that pharmacists were not well represented, as compared to doctors and nurses, because of their low staff strength in the institutions selected as study sites. To estimate the size of statistical effects when comparing characteristics between HCPs, it will be beneficial to recruit more pharmacists in future studies, in order to make an informed decision based on observed differences between these professionals. Moreover, the NH had a very small sample size, which may have also affected the statistical significance of tests of association associated with that facility.

## 5. Conclusion

There was a general awareness of AEFIs among HCPs in this study. However, negative perceptions and the lack of HCPs with high knowledge regarding AEFIs were possible setbacks to AEFI diagnosis, management, prevention, and reporting. More training and sensitization of HCPs on AEFIs and vaccine safety will be beneficial in improving the situation. Future research should focus on assessing the training materials and methodology used in informing HCPs about AEFIs and vaccine safety.

## Figures and Tables

**Table 1 vaccines-07-00028-t001:** Total cadres of each of the healthcare professionals against the desired sample sizes in the various hospitals.

Hospital	Doctors		Pharmacists		Nurses	
	Total	Needed	Total	Needed	Total	Needed
KATH	400	97	60	15	740	180
APH	96	59	3	3	139	86
TGH	120	52	10	6	370	159
NH	4	3	1	1	35	25
Total	620	211	74	25	1284	450

KATH: the Komfo Anokye Teaching Hospital; APH: the Agogo Presbyterian Hospital; TGH: the Tema General Hospital; NH: the Nigale Hospital.

**Table 2 vaccines-07-00028-t002:** Socio-demographic characteristics of study participants.

Classification	Variable, *n* = 453	Frequency (*n*)	Percentage (%)
Study location	Komfo Anokye Teaching Hospital	198	43.7
	Agogo Presbyterian Hospital	121	26.7
	Tema General Hospital	110	24.3
	Nigale Memorial Hospital	24	5.3
Age (years)	20–29	287	63.4
	30–39	144	31.8
	40–49	16	3.5
	≥50	6	1.3
Gender	Male	180	39.7
	Female	273	60.3
Profession	Doctor	121	26.7
	Pharmacist	25	5.5
	Nurse	307	67.8
Rank	Junior (1–5 years’ experience)	243	53.6
	Senior (≥5 years, but neither specialist nor consultant)	187	41.3
	Specialist	19	4.2
	Consultant	4	0.9
Area of practice	Immunization clinic	66	14.6
	Non-immunization clinic	387	85.4

**Table 3 vaccines-07-00028-t003:** Knowledge of study participants of AEFI.

Variable, *n* = 453	Frequency (*n*) *	Percentage (%) *
**Definition of AEFI acronym**		
Adverse Events Following Injection	54	11.9
Adverse Events Following Immunization	378	83.4
Abscess Following Immunization	17	3.8
Diseases with outbreak potential	2	0.4
Never heard about it	2	0.4
**Detailed knowledge of AEFI**		
AEFI is not limited to vaccination only.	227	50.1
AEFI can be caused by reconstituted vaccine stored longer than normal, vaccine reaction, inappropriate route of administration, vaccines stored beyond expiry date or contaminated vaccines.	317	70.0
Skin at injection site should be stretched during IM injection.	124	27.4
Paracetamol and ibuprofen are not used routinely to prevent fever before immunization.	165	36.4
Adrenaline should not be administered by SC route during anaphylaxis following immunization.	216	47.7
Investigation of an AEFI should commence within 24 h.	282	62.3
All injection site abscesses should be reported.	319	70.4
Injection site swelling and redness should be reported.	263	58.1
Treatment of a coincidental illness falsely attributed as a vaccine reaction should not be delayed until investigations are confirmed.	299	66.0
**AEFI knowledge categories**		
High knowledge (8–9)	49	10.8
Moderate knowledge (5–7)	213	47.0
Low knowledge (<5)	191	42.2
**AEFI knowledge acquisition at school, *n* = 453**		
Yes	76	16.7
No	377	83.2
**Number of AEFI trainings in the past one year, *n* = 453**		
None	357	78.8
1	53	11.6
2	28	6.2
3	9	2.0
More than 3	6	1.3

* Frequencies for detailed knowledge presented here are the ‘yes’ and correct answers for the various questions, whereas the percentages are for the number obtaining a correct mark out of all study participants.

**Table 4 vaccines-07-00028-t004:** Perceptions of AEFI among healthcare professionals.

Variable, *n* = 453	Yes (%) *	No (%) *
Believes reporting an AEFI cannot lead to personal consequences/punishment	115 (25.4)	338 (74.6)
Believes that reporting an AEFI will not make him/her feel guilty about having caused harm to a vaccinee	309 (68.2)	144 (31.8)
Believes that HCPs are willing to report an AEFI even when they are not confident about the diagnosis	226 (49.9)	227 (50.1)
Believes that poor monitoring of adverse events can cause reduction of immunization coverage	241 (53.2)	212 (46.8)
Believes that the process of reporting an AEFI is not long and tedious	268 (59.2)	185 (40.8)
Believes that if adverse events are reported, something will be done about it	235 (51.9)	218 (48.1)
Believes that enhancing surveillance of AEFI can help build public trust in immunization program	329 (72.6)	124 (27.4)
Desires to learn more about how to diagnose, report, investigate and manage AEFI	106 (23.4)	347 (76.6)
Believes he/she is busy but can still report AEFI	201 (44.4)	252 (55.6)
Believes he/she is interested in investigating or reporting AEFI	316 (69.8)	137 (30.2)

* ‘Yes’ answers signify positive perception whereas ‘No’ answers signify negative perception.

**Table 5 vaccines-07-00028-t005:** Tests of association between AEFI knowledge and study participant characteristics.

Classification	Variable	Knowledge of AEFI			df	*χ*^2^	*p*-value
		High	Moderate	Low			
Work location	KATH	24	96	78	9	16.017	<0.0001
	APH	18	54	49			
	TGH	7	52	51			
	NH	0	11	13			
Age	20–29	22	149	116	12	14.982	0.242
	30–39	18	57	69			
	40–49	6	5	5			
	≥50	3	2	1			
Gender	Male	12	51	117	9	20.645	0.006
	Female	37	162	74			
Profession	Doctor	19	61	41	9	13.384	0.146
	Pharmacist	3	14	8			
	Nurse	27	138	142			
Rank	Junior	11	143	89	15	15.405	0.423
	Senior	27	62	98			
	Specialist	9	6	4			
	Consultant	2	2	0			
Area of practice	Immunization clinic	37	18	11	6	6.703	0.349
	Non-immunization clinic	12	195	180			
AEFI acquisition from school	Yes	36	25	15	12	12.864	0.372
	No	13	188	176			
Number of AEFI trainings in past year	None	10	176	171	9	11.329	0.002
	1	16	26	11			
	2	14	8	6			
	3	5	1	3			
	More than 3	4	2	0			

KATH: the Komfo Anokye Teaching Hospital; APH: the Agogo Presbyterian Hospital; TGH: the Tema General Hospital; NH: the Nigale Hospital.

**Table 6 vaccines-07-00028-t006:** Summary of binary logistic regression of good perception versus participant characteristics.

Classification	Variable	Wald	OR (95% CI)	*p*-value
Work location	NH	Reference		
	KATH	6.11	2.73 (2.14–3.73)	0.002
	APH	10.64	5.28 (4.27–5.94)	<0.0001
	TGH	1.24	1.34 (1.02–1.78)	0.791
Age (years)	20–29	Reference		
	30–39	1.13	1.57 (1.19–2.68)	0.529
	40–49	2.76	1.09 (0.42–1.95)	0.081
	≥50	8.32	2.63 (1.92–3.54)	0.247
Gender	Male	Reference		
	Female	4.68	2.35 (2.03–3.42)	0.006
Profession	Doctor	Reference		
	Pharmacist	1.26	2.03 (1.26–2.49)	0.382
	Nurse	5.93	1.27 (0.71–1.84)	0.164
Rank	Junior	Reference		
	Senior	7.34	3.19 (2.38–3.16)	0.001
	Specialist	5.17	1.82 (1.13–2.24)	0.263
	Consultant	10.62	1.05 (0.52–1.79)	0.072
Area of practice	Non-immunization clinic	Reference		
	Immunization clinic	1.98	7.19 (6.25–7.81)	<0.0001
AEFI knowledge acquisition from school	No	Reference		
	Yes	3.41	5.07 (4.39–5.72)	0.017
Number of AEFI trainings in the past one year	None	Reference		
	1	17.28	4.92 (4.23–5.69)	0.028
	2	22.36	6.83 (6.16–7.62)	0.001
	3	1.53	2.1 (1.04–2.36)	0.319
	More than 3	3.07	4.32 (3.41–4.73)	0.087

KATH: the Komfo Anokye Teaching Hospital; APH: the Agogo Presbyterian Hospital; TGH: the Tema General Hospital; NH: the Nigale Hospital. Data shown are odds ratios (OR) of variables with 95% confidence interval (CI).

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
