# Peer review of "Knowledge and Perceptions of Adverse Events Following Immunization among Healthcare Professionals in Africa: A Case Study from Ghana"

_vaccines, 2019, doi:10.3390/vaccines7010028_

Round 1
Reviewer 1 Report
Dear Authors,
the manuscript focuses on a very important and current topic, the safety of vaccines. I red your manuscript with interest. However, in my opinion, the following minor revisions are required.
Introduction:
pag.1 line 38-40: Please add more elements and information to better define the context in low-income countries. Readers may not have deep knowledge of these countries and they could be helped in understanding the value of the study.
page 2, line 51-71: If i understand correctly, authors want to stress the importance of competence and techniques in delivering vaccines among healthcare professionals. I'm agree with that, but i suggest to shorten this part and if ever implement it with more refenrences.
page 2, line 72-75: This is a very intersting topic that it is possibile to observe in many fields of healthcare, included medical liability. I suggest to implement it, adding two-three sentences.
Discussion/Conclusion
Results about the relationship between the highest negative perception expressed by the lack of desire to learn more about how to diagnose report investigate and manage adaverse events following immunisation and the lowest with the lack of belief in survellaince improving public trust in immunisation programs is very interesting. Probably, i would have appreciated more about this correlation of results in the discussion.
Author Response
Dear Authors,
the
manuscript focuses on a very important and current topic, the safety of
vaccines. I red your manuscript with interest. However, in my opinion,
the following minor revisions are required.
Introduction:
pag.1
line 38-40: Please add more elements and information to better define
the context in low-income countries. Readers may not have deep knowledge
of these countries and they could be helped in understanding the value
of the study.
Response: The context in low income countries has been addressed by including the extent of the problem of low reporting of AEFIs and the fact that just about 10 countries (names mentioned in article) contribute to all AEFIs from Africa to the WHO medicine safety database. With the increasing doses of vaccines administered in Africa because of the burgeoning population and emergence of infectious diseases like Ebola virus disease, more AEFIs are expected and this must reflect in the quantity of AEFI reports submitted by Africa.
page 2,
line 51-71: If i understand correctly, authors want to stress the
importance of competence and techniques in delivering vaccines among
healthcare professionals. I'm agree with that, but i suggest to shorten
this part and if ever implement it with more refenrences.
Response: More literature from other jurisdictions like the UK have been included to broaden the scope of competency in delivering vaccines.
page
2, line 72-75: This is a very intersting topic that it is possibile to
observe in many fields of healthcare, included medical liability. I
suggest to implement it, adding two-three sentences.
Response: This portion has been modified to include lack of knowledge and poor perceptions and negligence in other healthcare fields such as radiology and its attendant problems leading to medical liability.
Discussion/Conclusion
Results about the relationship between the highest negative perception expressed by the lack of desire to learn more about how to diagnose report investigate and manage adaverse events following immunisation and the lowest with the lack of belief in survellaince improving public trust in immunisation programs is very interesting. Probably, i would have appreciated more about this correlation of results in the discussion.
Response 4: This has been addressed to some extent. Thank you.
Reviewer 2 Report
Thank you for the opportunity to review your interesting work. Please find below some suggestions as to how it could be enhanced:
Abstract:
Methods:You could add more specific methodology. For example, saying it occurred from 1st of July 2017, to the 31st of December, 2017 isn’t as meaningful as knowing whether you sent out reminders during this time to enhance the response rate. Mention that data was collected by means of a questionnaire when outlining study design (the only time questionnaire is mentioned is in the context of the number who returned it). Was it a paper-based questionnaire or electronic/online? Was it posted or hand-delivered and collected? How many questions in total? There should be some mention of data analysis as part of the methodology.
Results: I am not sure readers will appreciate what this statement really means if they just read the abstract: “knowledge of AEFIs was high 20 in 49 (10.8%) participants, moderate in 213 (47.0%) participants and low in 191 (42.2%) participants” as ‘high’, ‘moderate’ and ‘low’ haven’t previously been defined. Were there any differences in knowledge or perceptions between the three healthcare professionals i.e. doctors/pharmacists/nurses?
Discussion/Conclusion: If the last sentence in the abstract is meant to be a conclusion/discussion, then it is difficult to see how it’s been derived. None of these factors like ‘male gender’, ‘junior ranks’ have previously been mentioned in the context of the main findings, so it will not be apparent to the reader as to why you think they ‘may need more sensitization to improve their positive perceptions about AEFIs.’
Main paper:
Introduction:
Two sentences are very similar: “HCPs are a critical mass essential for developing post market surveillance systems in enhancing vaccine safety” and “Healthcare professionals (HCPs) form a very significant component of stakeholders of vaccine safety.” (lines 31-32 and 37-38)
Some of the introduction doesn’t necessarily help readers to appreciate why knowledge and perceptions of adverse events following immunization is vital (and hence why your work was important to undertake). i.e. having knowledge of vaccine storage and correct methods of administration is clearly important, but there needs to be clearer linkage to adverse events.
Can you provide a bit more information about the types of vaccinations that are routinely given so that the international reader-base can appreciate what they are and make comparisons as necessary? For example, the UK has an adult and child immunisation schedule with numerous recommended vaccinations - some are new, others well-established, some are associated with more risks than others or have been the subject of controversy etc.
Lines 62-65: When you are mentioning about stopping certain practices which also reduce the efficacy of vaccines (“analgesics around the time of administration of vaccine products”), you might want to mention the exception/Men B vaccine advice at 2 and 4 months of age (UK) where it is recommended to give paracetamol soon after vaccination, and not wait for a fever to develop. https://www.gov.uk/government/publications/menb-vaccine-and-paracetamol
(also, again – is this really helping to put your work into context for the reader – does reduced efficacy lead to increased likelihood of an adverse effect?)
Line 89 - change ‘adverse events following immunizations’ to AEFI
Lines 94-96 - Aim: I think this could be reworded/softened: you aimed to assess knowledge and perceptions in Ghana (with the hope this would inform practice elsewhere in Africa), rather than you aimed to assess knowledge and perceptions in Africa. You could also clarify who the healthcare professionals were in the aim/objectives.
Study design - perhaps it could encompass the word questionnaire - a cross-sectional questionnaire study or a cross-sectional quantitative study.
Why four hospitals (and why two in the north and two in the south rather than one in the north, south, east and west)?
Lines 116-117: You state that “Subjects in this study were frontline clinical healthcare professionals including doctors, pharmacists and nurses in the selected hospitals.” – the word ‘including’ implies there were others, yet your exclusion criteria states “Other clinical healthcare professionals apart from these were excluded from the study.” I suggest you change the first sentence to “Subjects in this study were frontline clinical healthcare professionals namely (or i.e.) doctors, pharmacists and nurses...”
Maybe you could clarify what ‘frontline’ means (I appreciate they may all have direct interactions with patients but do all of these healthcare professionals also administer vaccines?).
Line 131 – I think you need to rephrase this “…211 doctors, 25 pharmacists and 450 nurses were obtained for inclusion in the study.” to let readers know that this was the desired sample size otherwise it seems like you ‘obtained’ pharmacists, doctors and nurses.
Lines 135-142 – data collection tool. I think you could expand on development - whether there was reference to the wider literature in addition to the experts you’ve mentioned, the total number of questions in the questionnaire or the number per section, types of questions used (rating, ranking, open response etc.) paper or electronic and ways you tried to maximise the response rate from the outset.
Line 152-157 – Data collection - “Further explanations on study rationale, risks and benefits were given to participants who requested for it.” Remove the word ‘for’ within this sentence. Also, does this mean that you only provided this information to those who specifically requested it, rather than providing it to all participants automatically (as an information or cover sheet)? I don’t think it is clear to readers how the data collection actually occurred. Did this one assigned research assistant in each of the hospitals go round and manually distribute questionnaires to each potential participant and then collect them in again a time later (was it essentially self-completed by the participant) or did the research assistant help them to complete the questionnaires by providing explanations and clarifying queries etc.? Did the research assistant distribute consent forms in advance/how long was there between consenting to participate and participating?
Lines 164-168 - Knowledge levels and scores – had these been used previously in the literature or just something you assigned yourselves?
Table 2 – it might be helpful to clarify the difference between a specialist and a consultant and to be more explicit about the junior and senior categories in the context of both consultants and specialists. Were all consultants counted within the sole category ‘consultant’ regardless of number of years of experience (and the same for the specialists)?
Line 213 – Table 4 rather than table 4 (and for the others too)
Table 4 – some of these could be difficult to answer because they are asking more than one thing within the question. Also, readers might prefer to have the statements summarised in the positive rather than having to interpret a double negative “Does not believe…nothing will be done about it”. Ultimately, it’s more work for the reader to have ‘believe’ and ‘does not believe’ coupled with ‘yes’ and ‘no’ so you could re-do the statements (example wording shown below) which might have more impact in terms of helping the reader to quickly appreciate your key findings, while not changing the meaning of your results:
· XX% considered that enhancing surveillance of AEFI helped build public trust in immunization program
· XX% believed that poor monitoring of adverse events could cause a reduction of immunization coverage
· XX% thought that he/she is always busy and therefore could not report AEFI
· XX% deemed that the process of reporting an AEFI to be long and tedious
Lines 299-300: You state “A major strength of this study is the fact that the study sites consisted of primary, secondary and tertiary care facilities, making the sample representative.” Was it clear from the outset that this was the case? Also, the response rate could have been higher/mention possibility of non-response bias as a limitation. You could also mention more about the demographics in terms of limitations (for example, only 25 pharmacists participated).
Conclusion: a lot of this is repetition of the key findings rather than concluding remarks about the work/implications for practice/where future research should focus.
Author Response
Thank you for the opportunity to review your interesting work. Please find below some suggestions as to how it could be enhanced:
Abstract:
Methods:You could add more specific methodology. For example, saying it occurred from 1st of July 2017, to the 31st of December, 2017 isn’t as meaningful as knowing whether you sent out reminders during this time to enhance the response rate. Mention that data was collected by means of a questionnaire when outlining study design (the only time questionnaire is mentioned is in the context of the number who returned it). Was it a paper-based questionnaire or electronic/online? Was it posted or hand-delivered and collected? How many questions in total? There should be some mention of data analysis as part of the methodology.
Results: I am not sure readers will appreciate what this statement really means if they just read the abstract: “knowledge of AEFIs was high 20 in 49 (10.8%) participants, moderate in 213 (47.0%) participants and low in 191 (42.2%) participants” as ‘high’, ‘moderate’ and ‘low’ haven’t previously been defined. Were there any differences in knowledge or perceptions between the three healthcare professionals i.e. doctors/pharmacists/nurses?
Discussion/Conclusion: If the last sentence in the abstract is meant to be a conclusion/discussion, then it is difficult to see how it’s been derived. None of these factors like ‘male gender’, ‘junior ranks’ have previously been mentioned in the context of the main findings, so it will not be apparent to the reader as to why you think they ‘may need more sensitization to improve their positive perceptions about AEFIs.’
Response: Abstract expanded at methodology and results sections to make it flow logically. The conclusion has been summarized to address discussed results only.
Main paper:
Introduction:
Two sentences are very similar: “HCPs are a critical mass essential for developing post market surveillance systems in enhancing vaccine safety” and “Healthcare professionals (HCPs) form a very significant component of stakeholders of vaccine safety.” (lines 31-32 and 37-38)
Response: Line 37 and 38 have been deleted.
Some of the introduction doesn’t necessarily help readers to appreciate why knowledge and perceptions of adverse events following immunization is vital (and hence why your work was important to undertake). i.e. having knowledge of vaccine storage and correct methods of administration is clearly important, but there needs to be clearer linkage to adverse events.
Response: Poor cold chain storage linkage to AEFIs has been included and described. Temperature fluctuations could lead to the conversion of vaccine products to toxic substances which could trigger adverse events.
Can you provide a bit more information about the types of vaccinations that are routinely given so that the international reader-base can appreciate what they are and make comparisons as necessary? For example, the UK has an adult and child immunisation schedule with numerous recommended vaccinations - some are new, others well-established, some are associated with more risks than others or have been the subject of controversy etc.
Response: Types of vaccines used in children below 5 years in both African and European settings have been included at the introductory part of the manuscript.
Lines 62-65: When you are mentioning about stopping certain practices which also reduce the efficacy of vaccines (“analgesics around the time of administration of vaccine products”), you might want to mention the exception/Men B vaccine advice at 2 and 4 months of age (UK) where it is recommended to give paracetamol soon after vaccination, and not wait for a fever to develop. https://www.gov.uk/government/publications/menb-vaccine-and-paracetamol (also, again – is this really helping to put your work into context for the reader – does reduced efficacy lead to increased likelihood of an adverse effect?)
Response: The inclusion of Paracetamol and Ibuprofen being given to patients before immunization in the introduction of the write-up is a negative perception of importance which I thought had to be discussed because it is likely to let HCPs gloss of manifesting AEFIs which should be reported. Additionally, its effect on reduced immunogenicity of vaccines was described. The exception in Men B vaccine has been included.
Line 89 - change ‘adverse events following immunizations’ to AEFI
Response: Done.
Lines 94-96 - Aim: I think this could be reworded/softened: you aimed to assess knowledge and perceptions in Ghana (with the hope this would inform practice elsewhere in Africa), rather than you aimed to assess knowledge and perceptions in Africa. You could also clarify who the healthcare professionals were in the aim/objectives.
Response: Done.
Study design - perhaps it could encompass the word questionnaire - a cross-sectional questionnaire study or a cross-sectional quantitative study.
Response: Cross-sectional quantitative study has been implemented.
Why four hospitals (and why two in the north and two in the south rather than one in the north, south, east and west)?
Response: The hospitals were selected based on the needed representation of primary, secondary and tertiary levels and so a minimum of 3 with one from each category was needed. However, study was designed in such a way that there was one big hospital and one small hospital from each of the sectors and therefore, 4 was needed. North and South because that’s how healthcare facilities in Ghana have been categorized. The east, west are all part of southern sector. Remember, for the southern hospitals, one was from the western region and one from greater Accra region which is close to eastern region.
Lines 116-117: You state that “Subjects in this study were frontline clinical healthcare professionals including doctors, pharmacists and nurses in the selected hospitals.” – the word ‘including’ implies there were others, yet your exclusion criteria states “Other clinical healthcare professionals apart from these were excluded from the study.” I suggest you change the first sentence to “Subjects in this study were frontline clinical healthcare professionals namely (or i.e.) doctors, pharmacists and nurses...”
Response: Done.
Maybe you could clarify what ‘frontline’ means (I appreciate they may all have direct interactions with patients but do all of these healthcare professionals also administer vaccines?).
Response: Done.
Line 131 – I think you need to rephrase this “…211 doctors, 25 pharmacists and 450 nurses were obtained for inclusion in the study.” to let readers know that this was the desired sample size otherwise it seems like you ‘obtained’ pharmacists, doctors and nurses.
Response: Done.
Lines 135-142 – data collection tool. I think you could expand on development - whether there was reference to the wider literature in addition to the experts you’ve mentioned, the total number of questions in the questionnaire or the number per section, types of questions used (rating, ranking, open response etc.) paper or electronic and ways you tried to maximise the response rate from the outset.
Response: Data collection tool expanded.
Line 152-157 – Data collection - “Further explanations on study rationale, risks and benefits were given to participants who requested for it.” Remove the word ‘for’ within this sentence. Also, does this mean that you only provided this information to those who specifically requested it, rather than providing it to all participants automatically (as an information or cover sheet)? I don’t think it is clear to readers how the data collection actually occurred. Did this one assigned research assistant in each of the hospitals go round and manually distribute questionnaires to each potential participant and then collect them in again a time later (was it essentially self-completed by the participant) or did the research assistant help them to complete the questionnaires by providing explanations and clarifying queries etc.? Did the research assistant distribute consent forms in advance/how long was there between consenting to participate and participating?
Responses: An all-encompassing description has been given and details of data collection outlined.
Lines 164-168 - Knowledge levels and scores – had these been used previously in the literature or just something you assigned yourselves?
Response: Not previously done. Assigned by principal investigator and other vaccine pharmacovigilance experts
Table 2 – it might be helpful to clarify the difference between a specialist and a consultant and to be more explicit about the junior and senior categories in the context of both consultants and specialists. Were all consultants counted within the sole category ‘consultant’ regardless of number of years of experience (and the same for the specialists)?
Response: Description given.
Line 213 – Table 4 rather than table 4 (and for the others too)
Response: Done.
Table
4 – some of these could be difficult to answer because they are asking
more than one thing within the question. Also, readers might prefer to
have the statements summarised in the positive rather than having to
interpret a double negative “Does not believe…nothing will be done about
it”. Ultimately, it’s more work for the reader to have ‘believe’ and
‘does not believe’ coupled with ‘yes’ and ‘no’ so you could re-do the
statements (example wording shown below) which might have more impact in
terms of helping the reader to quickly appreciate your key findings,
while not changing the meaning of your results: Response: Simplified into simple positive terms. Double and triple negatives removed. · XX% considered that enhancing surveillance of AEFI helped build public trust in immunization program · XX% believed that poor monitoring of adverse events could cause a reduction of immunization coverage · XX% thought that he/she is always busy and therefore could not report AEFI · XX% deemed that the process of reporting an AEFI to be long and tedious Response: I tried this and it change the meaning of the results so I chose to rather avoid the double and triple negatives. Thanks. Lines
299-300: You state “A major strength of this study is the fact that the
study sites consisted of primary, secondary and tertiary care
facilities, making the sample representative.” Was it clear from the
outset that this was the case? Also, the response rate could have been
higher/mention possibility of non-response bias as a limitation. You
could also mention more about the demographics in terms of limitations
(for example, only 25 pharmacists participated). Response: Primary, secondary and tertiary now described in methodology. Conclusion: a lot
of this is repetition of the key findings rather than concluding
remarks about the work/implications for practice/where future research
should focus. Response: Regurgitated results removed and replaced with a focused conclusory message.
Reviewer 3 Report
This is a very interesting paper on an important topic. Ghana already has a good reputation with regard to pharmacovigilance and this research will hopefully encourage others to think more closely about how AEFIs are monitored and the support available for potential reporters.
The introduction set the scene well. There were 1 or 2 minor clarifications re English language clarity (e.g. needs 'where' after facilities on line 46; needs 'to' after according on line 69) but it was generally easy to follow and covered the key areas. I did however find the final objective to be very subjective with regard to the use of "good perception(s)" as a term. What is a 'good' perception? Can this be phrased more clearly / objectively?
The methods section needed a bit more information in a few areas
In section 2.2 it is stated that 4 hospitals are used as the study setting but the discussion (line 299-300) states that primary, secondary and tertiary facilities were included. This is not clear from the methods section.
With regard to the sampling, while the process for calculating sample sizes is very clearly described what is not clear is how those 686 individuals were recruited - did you select the individuals at random from the lists of staff or did you recruit sequentially on site until you had reached your numbers? This needs to be explained.
A few minor points in methods: Lines 125-126 - not sure why one hospital is abbreviated but the others are not - the abbreviated names have already been defined so no problem to abbreviate them all (consistency). Line 160 - statistics WERE (not was). Line 164-166 sentence is not very clear in terms of English.
With regard to the results, in table 3 the knowledge questions are not clearly presented - it would be helpful to distinguish between separate questions more clearly and to specify what the correct answer was. I assume this is how the questions were presented / worded to the respondents? It would also be better in tables to specify what 'percentage' means i.e. was it % correct, % who said 'yes', or % who selected this option - this could be done within the table legend.
I found the perceptions section incredibly difficult to follow. This was true for both the text and Table 4, especially with so many double (or even triple) negatives. E.g. the next to last one in Table 4: 56% said "no" to "does not believe he/she is always busy and cannot therefore report AEFI" - I think this means the 56% think they are too busy to report but even after re-reading several times I am still confused! It was also not clear which of the statements in the table were negative and which were positive as presented in the text (lines 208-214) due to all the double negatives / confusing wording.Could it be simplified in some way to make these results more accessible to readers? How were these questions presented to the respondents? Was the wording the same as in the table?
Table 5 highlights some differences based on characteristics using Chi squared test. However where the variables are not simply binary, it is not possible to see exactly where that difference lies (e.g. for number of training sessions). Some follow up statistical tests to explore this further would be of benefit.
Discussion: care re interpretation of results e.g. line 293-295 - higher education is associated with more positive perceptions but only up to a point - beyond 2 training sessions there is no significant difference. Likewise with ranks, it is only senior to junior that shows a difference - specialists and consultants did not show an increased positive attitude. Line 245: is the word 'reposes' what was intended here? It doesn't make sense to me.
Conclusion Would benefit from a rewrite to really make the take home messages sing out. At present , although it relates to the aims, it feels like hard work to pull out the main message - some restructuring / rephrasing may help it to stand out more.
Watch out for apostrophe usage throughout - some were missing from "participants" in some places
Author Response
This is a very interesting paper on an important topic. Ghana already has a good reputation with regard to pharmacovigilance and this research will hopefully encourage others to think more closely about how AEFIs are monitored and the support available for potential reporters.
The introduction set the scene well. There were 1 or 2 minor clarifications re English language clarity (e.g. needs 'where' after facilities on line 46; needs 'to' after according on line 69) but it was generally easy to follow and covered the key areas. I did however find the final objective to be very subjective with regard to the use of "good perception(s)" as a term. What is a 'good' perception? Can this be phrased more clearly / objectively?
Response: Objection rephrased.
The methods section needed a bit more information in a few areas
In section 2.2 it is stated that 4 hospitals are used as the study setting but the discussion (line 299-300) states that primary, secondary and tertiary facilities were included. This is not clear from the methods section.
Response: Done. Primary, secondary and tertiary described in methodology.
With regard to the sampling, while the process for
calculating sample sizes is very clearly described what is not clear is
how those 686 individuals were recruited - did you select the
individuals at random from the lists of staff or did you recruit
sequentially on site until you had reached your numbers? This needs to
be explained.
Response: Recruited sequentially until desired sample size was reached. Done.
A few minor points in methods: Lines 125-126 - not sure why one hospital is abbreviated but the others are not - the abbreviated names have already been defined so no problem to abbreviate them all (consistency). Line 160 - statistics WERE (not was). Line 164-166 sentence is not very clear in terms of English.
Response: Done.
With regard to the results, in table 3 the knowledge questions are not clearly presented - it would be helpful to distinguish between separate questions more clearly and to specify what the correct answer was. I assume this is how the questions were presented / worded to the respondents? It would also be better in tables to specify what 'percentage' means i.e. was it % correct, % who said 'yes', or % who selected this option - this could be done within the table legend.
Response: Done.
I found the perceptions section incredibly difficult to follow. This was true for both the text and Table 4, especially with so many double (or even triple) negatives. E.g. the next to last one in Table 4: 56% said "no" to "does not believe he/she is always busy and cannot therefore report AEFI" - I think this means the 56% think they are too busy to report but even after re-reading several times I am still confused! It was also not clear which of the statements in the table were negative and which were positive as presented in the text (lines 208-214) due to all the double negatives / confusing wording.Could it be simplified in some way to make these results more accessible to readers? How were these questions presented to the respondents? Was the wording the same as in the table?
Response: Double and triple negatives deleted and replaced with single positives.
Table 5 highlights some differences based on
characteristics using Chi squared test. However where the variables are
not simply binary, it is not possible to see exactly where that
difference lies (e.g. for number of training sessions). Some follow up
statistical tests to explore this further would be of benefit.
Response: The difference is there please. For instance for number of AEFI trainings in past year, the high knowledge scores shows 16 participants for 1 year, 14 for 2 years, 10 for none, 5 for 3 years and 4 for more than 3 years. So before the chi square test, this is the order regarding high knowledge. However, chi square says this order is not statistically significant. Similar tests with one way ANOVA and other tests using the mean yielded virtually the same results and I felt there is no need to repeat results.
Discussion: care re interpretation of results e.g. line 293-295 - higher education is associated with more positive perceptions but only up to a point - beyond 2 training sessions there is no significant difference. Likewise with ranks, it is only senior to junior that shows a difference - specialists and consultants did not show an increased positive attitude. Line 245: is the word 'reposes' what was intended here? It doesn't make sense to me.
Response: These have been addressed.
Conclusion Would benefit from a rewrite to really make the take home messages sing out. At present , although it relates to the aims, it feels like hard work to pull out the main message - some restructuring / rephrasing may help it to stand out more.
Response: Done.
Watch out for apostrophe usage throughout - some were missing from "participants" in some places.
Response: Noted.
Round 2
Reviewer 2 Report
I am satisfied that my comments have been addressed to a satisfactory standard and hope you agree this has improved the quality of the work.
I still think it would have been possible to present Table 4 results in bullet points as previously suggested without altering the meaning/essence of the findings. However, I will leave this at your discretion.